# HOW TO MODEL YOUR COVARIANCE

## ABSTRACT

We study the problem of *unsupervised heteroscedastic covariance estimation*, where we wish to learn the multivariate target distribution $\mathcal{N}(\boldsymbol{\mu}_{Y|X}, \boldsymbol{\Sigma}_{Y|X}|X{=}\boldsymbol{x})$ given an input $\boldsymbol{x}$. This problem is particularly challenging as $\boldsymbol{\Sigma}_{Y|X}$ varies for different samples (heteroscedastic) and no annotation for the covariance is available (unsupervised). Typically, state-of-the-art methods learn the mean and covariance of this distribution through two neural networks trained using the negative log-likelihood. This raises two questions: (1) Does the predicted covariance truly capture the randomness of the predicted mean? (2) In the absence of ground-truth annotation, how can we quantify the performance of covariance estimation? We address (1) by deriving the *Spatial Variance*, which captures the randomness of the multivariate prediction by incorporating its gradient and curvature through the second order Taylor polynomial. Furthermore, we tackle (2) by introducing the *Conditional Mean Absolute Error (CMAE)*, a metric which leverages conditioning of the normal distribution to evaluate the covariance. We verify the effectiveness of our approach through multiple experiments spanning synthetic (univariate, multivariate) and real-world datasets (UCI Regression, LSP, and MPII Human Pose). Our experiments show that the Spatial Variance outperforms state-of-the-art in accurately learning the covariance, as quantified through the CMAE.

## 1 INTRODUCTION

Table 1: *Notation*. Given samples $(\boldsymbol{x}, \boldsymbol{y})$ drawn from the unknown distribution $p(X, Y)$, our goal is to estimate the covariance of target distribution $p(Y|X = \boldsymbol{x}) = \mathcal{N}(\boldsymbol{\mu}_{Y|X}, \boldsymbol{\Sigma}_{Y|X}|X = \boldsymbol{x})$ given $\boldsymbol{x}$.

| Estimator | Input | Networks | Labels | Prediction | Target | Shape | Supervised? |
|---|---|---|---|---|---|---|---|
| Mean | $\boldsymbol{x}$ | $f_\theta$ | $\boldsymbol{y}$ | $\hat{\boldsymbol{y}}$ | $\boldsymbol{\mu}_{Y|X}$ | $n$ | **Yes** |
| Covariance | | $g_\Theta$ | - | $\mathrm{Cov}(\hat{Y}|X)$ | $\boldsymbol{\Sigma}_{Y|X}$ | $n \times n$ | **No** |

Modeling the target distribution $p(Y|X{=}\boldsymbol{x})$ is an important design choice in regression. The standard assumption is that the target follows a multivariate normal distribution $\mathcal{N}(\boldsymbol{\mu}_{Y|X}, \boldsymbol{\Sigma}_{Y|X} \mid X{=}\boldsymbol{x})$, where the true mean $\boldsymbol{\mu}_{Y|X}$ and covariance $\boldsymbol{\Sigma}_{Y|X}$ are unknown. *The challenge is that while estimating the mean is a supervised task, covariance estimation is unsupervised.* Indeed, while $(\boldsymbol{x}, \boldsymbol{y})$ are observed, $\boldsymbol{\Sigma}_{Y|X}$ needs to be inferred. Moreover, $\boldsymbol{\Sigma}_{Y|X}$ is often heteroscedastic and takes on different values for different input samples. Regressing $\boldsymbol{x}$ to $\boldsymbol{y}$ can be simplified by assuming the covariance to be an identity matrix, $\boldsymbol{\Sigma}_{Y|X} = \boldsymbol{I}_n$, or ignoring correlations ($\boldsymbol{\Sigma}_{Y|X} = \mathrm{diag}(\boldsymbol{\sigma})$). However, such approaches diminish the main advantages of learning the covariance, such as correlation analysis, sampling from the predicted distribution $q(\hat{Y}|X{=}\boldsymbol{x})$, and updating our predictions conditioned on partial observations of the target. Typically, covariance estimation is performed through minimizing the negative log-likelihood (NLL) of the predicted distribution $q(\hat{Y}|X{=}\boldsymbol{x}) = \mathcal{N}(\hat{\boldsymbol{y}}, \mathrm{Cov}(\hat{Y}|X))$. This involves the joint optimization of the predicted mean $\hat{\boldsymbol{y}} = f_\theta(\boldsymbol{x})$ and the covariance $\mathrm{Cov}(\hat{Y}|X) = g_\Theta(\boldsymbol{x})$ estimators (Dorta et al., 2018) over the dataset:

$$\mathbb{E}_{p(X,Y)}\left[-\log\, q_{\theta,\Theta}(\hat{Y}|X{=}\boldsymbol{x})\right] = \frac{1}{N}\sum_{(\boldsymbol{x},\boldsymbol{y})^{(1)}}^{(\boldsymbol{x},\boldsymbol{y})^N}\left[\log\left|\mathrm{Cov}(\hat{Y}|X)\right| + (\boldsymbol{y}{-}\hat{\boldsymbol{y}})^T\,\mathrm{Cov}(\hat{Y}|X)^{-1}\,(\boldsymbol{y}{-}\hat{\boldsymbol{y}})\right].$$

(1)

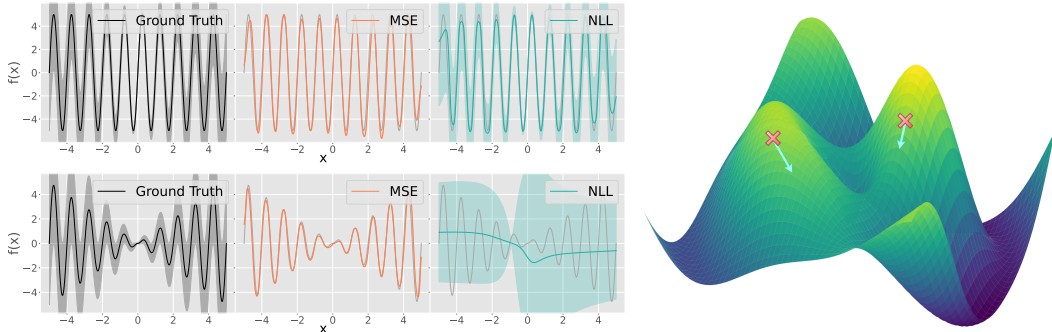

Figure 1: (Left) The network is trained to predict a constant and varying amplitude sinusoidal with heteroscedastic noise. We observe sub-optimal convergence with negative log-likelihood because $\mathrm{Var}(\hat{Y}|X) = g_\Theta(\boldsymbol{x})$ may predict arbitrary values to minimize the objective. As a result, $\mathrm{Var}(\hat{Y}|X)$ may not truly explain the randomness in $\hat{\boldsymbol{y}}$. (Right) In contrast, *Spatial Variance* shows improved results by explaining the randomness in $\hat{\boldsymbol{y}}$ through its gradient and curvature. Intuitively, the gradient and curvature quantify the variation in the prediction within an $\epsilon$-neighborhood of $\boldsymbol{x}$.

However, state-of-the-art results (Seitzer et al., 2022; Stirn et al., 2023) show that this joint optimization of $f_\theta$ and $g_\Theta$ causes sub-optimal target $\hat{\boldsymbol{y}}$ predictions because of incorrect covariance predictions. *Indeed, we observe in Fig. 1 that without supervision, $g_\Theta(\boldsymbol{x})$ maps $\boldsymbol{x}$ to possibly arbitrary variances that minimize the objective. As a result, the predicted covariance may not truly explain the randomness in $\hat{\boldsymbol{y}}$.* Moreover, assessing the quality of covariance estimation is challenging without ground-truth labels. Optimization metrics such as likelihood scores are not a direct measure of the accuracy of the correlations learnt by the covariance estimator since they also incorporate the performance of the mean estimator. For instance, the mean square error would be zero for the perfect estimator $f_\theta(\boldsymbol{x})$, and completely disregard the covariance. Alternative measure such as log-likelihood put greater emphasis on the determinant of the covariance without directly assessing the covariance. Therefore, we distill the challenges associated with covariance estimation into two problem statements: (1) How do we formulate $\mathrm{Cov}(\hat{Y}|X)$ to explain the randomness in $\hat{\boldsymbol{y}}$? (2) How do we assess the quality of covariance estimation in the absence of ground-truth annotations?

Our first major contribution, the **Spatial Variance**, explains the randomness in $f_\theta(\boldsymbol{x})$ through a novel derivation of a closed form expression for the predicted covariance. Specifically, we derive this expression by solving for the covariance of the multivariate target $f_\theta(\boldsymbol{x} + \epsilon)$ through its Taylor expansion around $\boldsymbol{x}$. As a result we can model the covariance through the gradient and curvature of $f_\theta(\boldsymbol{x})$ which captures the variation in the prediction within an $\epsilon$-neighborhood of $\boldsymbol{x}$ (Fig. 1). We show that such a formulation when learnt through the negative log-likelihood outperforms other baselines.

Our second major contribution, the **Conditional Mean Absolute Error (CMAE)**, addresses the lack of a direct metric to estimate the quality of covariance estimation. By definition, an accurate covariance prediction correctly estimates the correlations underlying the target random variables. Hence, given a partial observation of the target $\boldsymbol{y}_{[1:k]}$, the covariance should accurately update the prediction $\tilde{\boldsymbol{y}}_{[k:n]}$ towards the unobserved target $\boldsymbol{y}_{[k:n]}$ based on conditioning of the target distribution $\mathcal{N}(\tilde{\boldsymbol{y}}_{[k:n]}, \mathrm{Cov}(\hat{Y}|X) \,|\, \boldsymbol{x}, \boldsymbol{y}_{[1:k]})$. Subsequently, we quantify CMAE as the mean absolute error between the updated prediction and the unobserved target.

We design and perform extensive experiments on synthetic (sinusoidal, multivariate) and real-world datasets (UCI Regression and Human Pose - MPII, LSP), across fully-connected and convolutional network architectures. Using CMAE, our experiments show that the Spatial Variance outperforms state-of-the-art baselines in learning correlations across all tasks and architectures, demonstrating the effectiveness of our method for unsupervised heteroscedastic covariance estimation.

## 2 RELATED WORK

**Negative Log-Likelihood Based Approaches**: Unsupervised covariance estimation is popularly done through negative log-likelihood based optimization (Dorta et al., 2018; Gundavarapu et al.,

2019; Russell & Reale, 2021; Lu & Koniusz, 2022; Simpson et al., 2022; Liu et al., 2018; Barratt & Boyd, 2022; Kendall & Gal, 2017). Recently, Seitzer et al. (2022) through the diagonal covariance $\Sigma_{Y|X} = \text{diag}(\boldsymbol{\sigma})$ show that the negative log-likelihood results in sub-optimal fits due to the incorrect scaling of the squared residual $(\boldsymbol{y} - f_\theta(\boldsymbol{x}))^2$ by the variance $\text{Var}(\hat{Y}|X) = g_\Theta(\boldsymbol{x})$. Subsequently, the work proposes $\beta$-NLL, which scales the negative log-likelihood objective with $\text{Var}(\hat{Y}|X)^\beta$, thereby reducing the impact of the predicted variance in the training process. Intuitively, any $\beta \neq \{0, 1\}$ provides a degree of trade-off between the negative log-likelihood and the mean squared error. While $\beta$-NLL is delightfully simple and effective, the method has key limitations. First, $\beta$-NLL is not a result of a valid distribution, and the optimized values do not translate to the variance of the distribution. Second, $\beta$-NLL is a scaled variant of the negative log-likelihood; samples that are inherently noisy take larger gradient steps in comparison to clean samples, leading to imbalanced convergence. The recent method of Stirn et al. (2023) proposes an alternative approach and allows the target estimator to converge faster by training it with identity covariance. However, this involves conflicting assumptions; while the target estimator assumes that variables within $\boldsymbol{y} - f_\theta(\boldsymbol{x})$ are uncorrelated, the covariance estimator is expected to recover correlations from the residual.

Finally, the drawback of $\text{Cov}(\hat{Y}|X) = g_\Theta(\boldsymbol{x})$ being an arbitrary mapping from $\boldsymbol{x}$ to a positive definite matrix is common to all the aforementioned approaches. This drawback is significant since in the absence of supervision, $g_\Theta(\boldsymbol{x})$ can take on any value which minimizes the objective and does not necessarily represent the randomness of the prediction. Therefore, we present a novel derivation of a closed form expression for the predicted covariance and show that incorporating the gradient and curvature better explains the randomness in $\hat{\boldsymbol{y}}$.

**Probabilistic Methods**: The statistical properties of covariance have been well explored in the community (Biswas et al., 2020; Hoff et al., 2022; Kastner, 2019; Nguyen et al., 2022; Chen et al., 2017), however, these methods do not extrapolate to cases where we wish to condition the target given an input image. Ensembles are useful in computing epistemic uncertainty Kendall & Gal (2017); Lakshminarayanan et al. (2017), however the method quantifies the disagreement over the target and does not represent the correlations within the target. Lotfi et al. (2022) proposes the Conditional Marginal Likelihood (CML) as a metric to improve generalization. The metric creates two subsets of the dataset, and quantifies the performance on the second subset conditioned on a model trained only on the first subset. In contrast, CMAE is a metric with a different objective of evaluating the covariance of the target prediction per samples of the dataset.

**Hessians in Deep Learning**: The Hessian is interpreted as the curvature of the loss landscape and is significant in optimization Gilmer et al. (2022). Approaches that study uncertainty and optimization using the parameters' gradients include Jacot et al. (2018); Van Amersfoort et al. (2020). The duality between gradient and Hessian has been well studied through the Fisher information (Ly et al., 2017) and applied in uncertainty estimation (Shukla, 2022). Of particular importance is the Cramer-Rao bound which is used in multiple optimizers such as Adam (Kingma & Ba, 2015). Classical (Kanazawa & Kanatani, 2003) and recent (Tirado-Garín et al., 2023) works in the domain of image processing use the Cramer-Rao bound to compute two dimensional covariances based on the heatmap of image descriptors. However, the Cramer-Rao bound estimates the variance of the parametric estimator, and cannot be used to estimate the covariance of the prediction. This limitation cannot be circumvented since Fisher information averages the network gradient over all samples, losing the ability to estimate the covariance for each sample. Therefore, we derive the covariance through a first principles' approach by attempting to solve for $\text{Cov}(\hat{Y}|X)$ directly.

## 3 Spatial Variance

We return to the prediction distribution $q(\hat{Y}|X{=}\boldsymbol{x})$ and ponder on a fundamental question: what is the randomness of a prediction $\hat{\boldsymbol{y}}$ for a sample $\boldsymbol{x}$?

### 3.1 $\epsilon$-Neighborhood

Let us assume we have $N$ predictions $(x, \hat{y})^{(1)} \dots (x, \hat{y})^{(N)}$. While, $\text{Cov}(\hat{Y}|X{=}x)$ is a measure of randomness, the probability of exactly observing $p(X = x)$ is zero for continuous variables. Instead, the standard approach (Evans & Rosenthal, 2004) is to observe over the set $X \in lim_{\varepsilon \to 0}[x, x + \varepsilon]$. This implies that for continuous variables, we do not observe a specific but a range of values around

$X = x$. We can incorporate this definition in the covariance too. Since $\hat{Y}$ is a deterministic transformation of $X$ through a parametric network, $\hat{Y} = f_\theta(X)$, we have

$$\text{Cov}(\hat{Y}|X=x) = \text{Cov}(f_\theta(X) \mid X \in lim_{\varepsilon \to 0}[x, x + \epsilon]) \ . \tag{2}$$

Intuitively, Eq. 2 allows us to represent the covariance as the change in $\hat{\boldsymbol{y}} = f_\theta(\boldsymbol{x})$ within an $\varepsilon$-neighborhood around $\boldsymbol{x}$. We go a step further and interpret this neighborhood $lim_{\varepsilon \to 0}[x, x + \varepsilon]$ as $x + \epsilon$, where $\epsilon$ is an m-dimensional random variable with a zero-mean isotropic Gaussian distribution $p(\boldsymbol{\epsilon}) = \mathcal{N}(0, \sigma_\epsilon^2(\boldsymbol{x})\boldsymbol{I}_m)$. Introducing $\epsilon$ imposes a distribution on the neighborhood which is centred at $\boldsymbol{x}$. This also allows us to represent $\hat{y} = f_\theta(\boldsymbol{x} + \epsilon)$ stochastically. While the variance of this random variable is unknown (we later show that it can be learnt), we assume heteroscedasticity which allows us to represent neighborhoods of varying spatial extents for each $\boldsymbol{x}$. Subsequently, Eq. 2 becomes

$$\text{Cov}(\hat{Y}|X=x) = \text{Cov}(f_\theta(x + \epsilon)) \ . \tag{3}$$

We continue our analysis by taking the Taylor expansion of $f_\theta(\boldsymbol{x} + \epsilon)$.

## 3.2 MULTIVARIATE TAYLOR EXPANSION

The Taylor expansion introduces the notion of gradient and curvature in modeling the covariance, and quantifies the rate at which a function can change within a small neighborhood around $\boldsymbol{x}$. The multivariate Taylor expansion in a matrix notation is given by

$$f_\theta(\boldsymbol{x} + \epsilon) \approx f_\theta(\boldsymbol{x}) + \boldsymbol{J}(\boldsymbol{x})\epsilon^T + \frac{\boldsymbol{h}}{2} \ , \qquad \text{where } \boldsymbol{h}_i = \epsilon\,\mathsf{H}_i(\boldsymbol{x})\epsilon^T \ \forall i \in 1 \dots n \ . \tag{4}$$

Here, $f_\theta(\boldsymbol{x}) \in \mathbb{R}^n$ represents the multivariate prediction, $\epsilon \in \mathbb{R}^m$ represents the neighborhood of $\boldsymbol{x}$, $\boldsymbol{J}(\boldsymbol{x}) \in \mathbb{R}^{n \times m}$ corresponds to the Jacobian matrix and $\mathsf{H}(\boldsymbol{x}) \in \mathbb{R}^{n \times m \times m}$ represents the Hessian tensor. Note that all the individual terms in Eq. 4 are $n$-dimensional.

## 3.3 COVARIANCE ESTIMATION

The covariance of Eq. 4 with respect to the random variable $\epsilon$ is given by

$$\text{Cov} f_\theta(\boldsymbol{x} + \epsilon) = \text{Cov}\left[ f_\theta(\boldsymbol{x}) + \boldsymbol{J}(\boldsymbol{x})\epsilon^T + \frac{\boldsymbol{h}}{2} \right]$$

$$= \text{Cov}(\boldsymbol{J}(\boldsymbol{x})\epsilon^T) + \text{Cov}(\frac{\boldsymbol{h}}{2}) + 2\left[ \text{Cov}(\boldsymbol{J}(\boldsymbol{x})\epsilon^T, \frac{\boldsymbol{h}}{2}) \right] \ . \tag{5}$$

**a. Estimating** $\text{Cov}(\boldsymbol{J}(\boldsymbol{x})\epsilon^T, \boldsymbol{h}/2)$. We begin by noting that $\boldsymbol{J}(\boldsymbol{x})\epsilon^T$ and $\boldsymbol{h}$ are $n$-dimensional vectors with elements $[\dots \boldsymbol{J}_i(\boldsymbol{x})\epsilon^T \dots]$ and $[\dots \epsilon\,\mathsf{H}_k(\boldsymbol{x})\epsilon^T \dots]$ respectively. The covariance between any two elements is given by

$$\text{Cov}\big(\boldsymbol{J}_i(\boldsymbol{x})\epsilon^T, \epsilon\,\mathsf{H}_k(\boldsymbol{x})\epsilon^T\big) = \mathbb{E}\big(\boldsymbol{J}_i(\boldsymbol{x})\epsilon^T \epsilon\,\mathsf{H}_k(\boldsymbol{x})\epsilon^T\big) - \mathbb{E}\big(\boldsymbol{J}_i(\boldsymbol{x})\epsilon^T\big)\mathbb{E}\big(\epsilon\,\mathsf{H}_k(\boldsymbol{x})\epsilon^T\big)$$

$$= 0 \ . \tag{6}$$

**Odd and Even Functions.** We use the property of odd-even functions (Shynk, 2012) to arrive at this solution. We recall that an odd function is defined as $f(-t) = -f(t)$ and an even function as $f(-t) = f(t)$. Next, we note that the product of an odd and even function is odd, and the product of two even functions is even. Finally, the integral of an odd function over its domain evaluates to zero.

We note that $\boldsymbol{J}_i(\boldsymbol{x})\epsilon^T = \sum_k \boldsymbol{J}_{i,k}(\boldsymbol{x})\epsilon_k^T$ as an odd function with respect to $\epsilon$. Further, our design choice of $p(\boldsymbol{\epsilon}) = \mathcal{N}(0, \sigma_\epsilon^2(\boldsymbol{x})\boldsymbol{I}_m)$ implies that the distribution $p(\boldsymbol{\epsilon})$ is an even function. The term $\mathbb{E}\big(\boldsymbol{J}_i(\boldsymbol{x})\epsilon^T\big)$ can be written as $\int_\epsilon \boldsymbol{J}_i(\boldsymbol{x})\epsilon^T p(\epsilon)\mathrm{d}\epsilon$. This term represents the integral of a product of an odd and even function, which evaluates to zero.

The quadratic term $\epsilon\,\mathsf{H}_k(\boldsymbol{x})\epsilon^T$ can be written as $\sum_i \sum_j \mathsf{H}_{i,j}^{(k)} \epsilon_i \epsilon_j$, which is an even function. Subsequently, $\boldsymbol{J}_i(\boldsymbol{x})\epsilon^T \epsilon\,\mathsf{H}_k(\boldsymbol{x})\epsilon^T$ is a product of odd $\boldsymbol{J}_i(\boldsymbol{x})\epsilon^T$ and even $\epsilon\,\mathsf{H}_k(\boldsymbol{x})\epsilon^T$ terms. Finally, we can write $\mathbb{E}\big(\boldsymbol{J}_i(\boldsymbol{x})\epsilon^T \epsilon\,\mathsf{H}_k(\boldsymbol{x})\epsilon^T\big)$ as $\int_\epsilon \boldsymbol{J}_i(\boldsymbol{x})\epsilon^T \epsilon\,\mathsf{H}_k(\boldsymbol{x})\epsilon^T p(\epsilon)\mathrm{d}\epsilon$, which represents the integral of a product of odd, even and even functions, which also evaluates to zero.

As a result, we get $\text{Cov}\big(\boldsymbol{J}_i(\boldsymbol{x})\epsilon^T, \epsilon\,\mathsf{H}_k(\boldsymbol{x})\epsilon^T\big) = 0 \ \forall i, k$, implying that $\text{Cov}(\boldsymbol{J}(\boldsymbol{x})\epsilon^T, \boldsymbol{h}/2) = 0$.

**Algorithm 1:** *Conditional Mean Absolute Error*

**Input:** $y$: Ground truth
**Input:** $\hat{y}$: Target prediction
**Input:** $\mathrm{Cov}(\hat{Y}|X)$: Covariance prediction
**Output:** CMAE: Conditional Mean Absolute Error

1 dimensions = `get_dimensions`($\hat{y}$)
2 CMAE = `zeros`(shape=dimensions)

3 **for** *i in dimensions* **do**

4      obs_dim = `set`(dimensions) - `set`(i)
5      hidden_dim = i

6      $\boldsymbol{\Sigma}_{22} = \mathrm{Cov}(\hat{Y}|X)$[obs_dim, obs_dim]
7      $\boldsymbol{\Sigma}_{12} = \mathrm{Cov}(\hat{Y}|X)$[hidden_dim, obs_dim]
8      $\tilde{\boldsymbol{y}} = \hat{\boldsymbol{y}}$[hidden_dim] $+ (\boldsymbol{\Sigma}_{12}\boldsymbol{\Sigma}_{22}^{-1}$
         $(\boldsymbol{y}$[obs_dim] $- \hat{\boldsymbol{y}}$[obs_dim]$))$

9      CMAE[i] = $|\tilde{\boldsymbol{y}} - \boldsymbol{y}$[hidden_dim]$|$

10 **return** CMAE.`mean()`

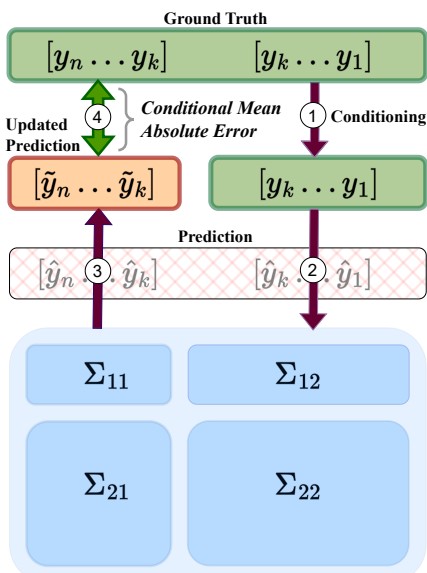

Figure 2: We propose the Conditional Mean Absolute Error (CMAE) as a metric for covariance estimation. Given the ground truths, predicted mean and the predicted covariance matrix, CMAE quantifies the improvement in the predicted mean given partial observations of the ground truth. CMAE uses conditioning of the normal distribution as a direct measure to assess the covariance.

***b. Estimating*** $\mathrm{Cov}(\boldsymbol{J}(\boldsymbol{x})\epsilon^T)$ ***and*** $\mathrm{Cov}(\boldsymbol{h}/2)$. Estimating $\mathrm{Cov}(\boldsymbol{J}(\boldsymbol{x})\epsilon^T)$ and $\mathrm{Cov}(\boldsymbol{h}/2)$ in Eq. 5 is easier since they follow a linear and quadratic form respectively with known solutions for isotropic Gaussian random variables (Eq. 375, 379 in (Petersen & Pedersen, 2012)). Specifically, we have

$$\mathrm{Cov}(\boldsymbol{J}(\boldsymbol{x})\ \epsilon^T) = k_1(x)\boldsymbol{J}(\boldsymbol{x})\boldsymbol{J}(\boldsymbol{x})^T \quad \text{and} \quad \mathrm{Cov}(\boldsymbol{h}/2)_{i,j} = k_2(x)\ \texttt{Trace}\ (\mathbf{H}_{i,:,:}(\boldsymbol{x})\ \mathbf{H}_{j,:,:}(\boldsymbol{x}))\ . \quad (7)$$

Since we do not know the variance of the $\epsilon$ and its transformation for each $\boldsymbol{x}$, we define them through positive quantities $k_1(\boldsymbol{x})$ and $k_2(\boldsymbol{x})$ which are optimized by the covariance estimator $g_\Theta(\boldsymbol{x})$. Finally, we note that both $\mathrm{Cov}(\boldsymbol{J}(\boldsymbol{x})\ \epsilon^T)$ and $\mathrm{Cov}(\boldsymbol{h}/2)$ have dimensions $n \times n$. We obtain the expression for the Spatial Variance by substituting Eq. 7 and Eq. 6 into Eq. 5:

$$\mathrm{Cov} f_\theta(\boldsymbol{x} + \epsilon) \approx k_1(\boldsymbol{x})\boldsymbol{J}(\boldsymbol{x})\boldsymbol{J}(\boldsymbol{x})^T + \mathcal{H} \quad \text{where} \quad \mathcal{H}_{i,j} = k_2(\boldsymbol{x})\ \texttt{Trace}\ (\mathbf{H}_{i,:,:}(\boldsymbol{x})\ \mathbf{H}_{j,:,:}(\boldsymbol{x}))\ . \quad (8)$$

### 3.4 FORMULATION

For highly stochastic samples, the gradient and curvature may not sufficiently explain the covariance. For example. the regressed line $f_\theta(x) = \boldsymbol{c}$ perfectly fits the function $\boldsymbol{y} = \boldsymbol{c} + \varepsilon$, where $\varepsilon \sim \mathcal{N}(0, \boldsymbol{\Sigma})$. However, the gradient and curvature of this function are zero and hence fail to capture $\boldsymbol{\Sigma}$ . Therefore, we remedy this by adding $k_3(\boldsymbol{x}) \in \mathbb{R}^{n \times n}$, a learnable positive definite matrix to Eq. 8 for the covariance estimator $g_\Theta(x)$ to optimize. The final expression for the Spatial Variance is

$$\mathrm{Cov}(\hat{Y}|X = x) \approx k_1(\boldsymbol{x})\boldsymbol{J}(\boldsymbol{x})\boldsymbol{J}(\boldsymbol{x})^T + \mathcal{H} + k_3(\boldsymbol{x})\ . \quad (9)$$

The covariance estimator $g_\Theta(x)$ predicts $k_1(\boldsymbol{x}), k_2(\boldsymbol{x})$ and $k_3(\boldsymbol{x})$, where $k_1(\boldsymbol{x}), k_2(\boldsymbol{x})$ are positive scalars. We enforce $k_3(x)$ to be positive semi-definite by predicting an unconstrained matrix and multiplying it with its transpose, similar to previous work. The covariance estimator is trained to minimize the negative log-likelihood by substituting Eq. 9 into Eq. 1. Intuitively, the advantage of Spatial Variance is that unlike the traditional method, we quantify the covariance as a function of how quickly the mean estimator changes within a small radius of $\boldsymbol{x}$. Larger derivatives imply a rapid change in $\boldsymbol{y}$ and as a result, the model has a higher variance about its estimate. With our experiments we show that incorporating Spatial Variance in negative log-likelihood results in significant improvements.

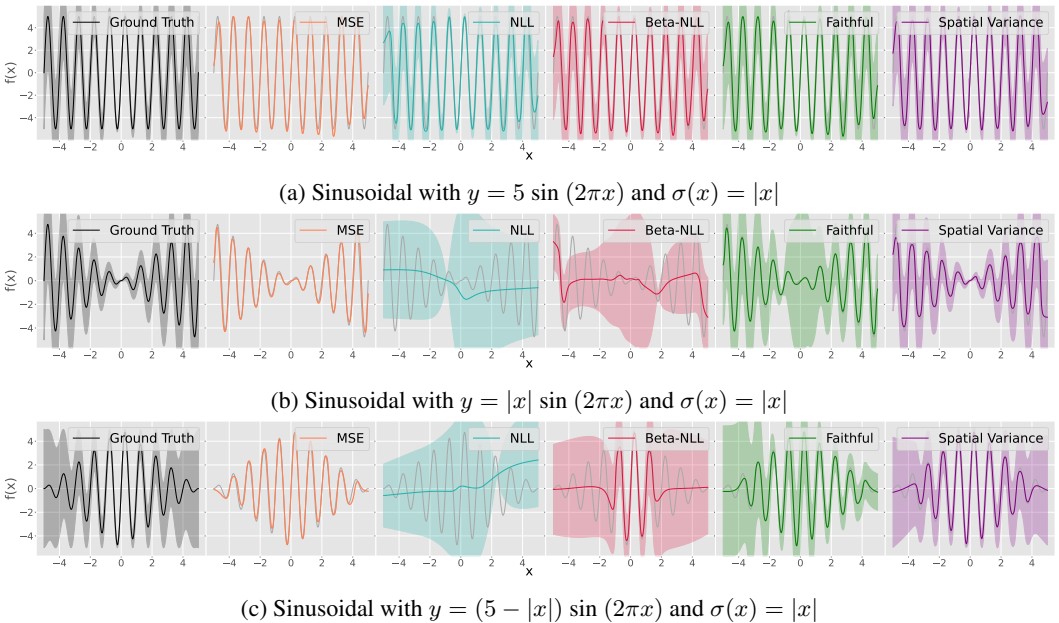

(a) Sinusoidal with $y = 5 \sin (2\pi x)$ and $\sigma(x) = |x|$

(b) Sinusoidal with $y = |x| \sin (2\pi x)$ and $\sigma(x) = |x|$

(c) Sinusoidal with $y = (5 - |x|) \sin (2\pi x)$ and $\sigma(x) = |x|$

Figure 3: *Synthetic-Univariate*. We show that the variances learnt by mapping may not explain the randomness in the model predictions. In comparison, the Spatial Variance performs better since it explains the (co-)variance of the predictions through the model gradient and curvature.

### 3.5 LIMITATIONS

The Spatial Variance is computationally expensive since the local curvature needs to be computed for each sample. While parallel computing is useful, the method is not real-time. In practice, covariance estimation can be performed using a smaller, proxy model in place of a large model (which could be retained for mean estimation). The reduced parameter count would decrease the computational requirements of computing the Hessian.

## 4 CMAE: CONDITIONAL MEAN ABSOLUTE ERROR

How can we quantify the accuracy of our covariance estimates in the absence of ground truth annotation? Existing techniques (Kendall & Gal, 2017; Seitzer et al., 2022; Stirn et al., 2023) use metrics such as likelihood scores and mean square error for evaluation. However, these methods are skewed towards learning the mean; a perfect estimator $f_\theta(x)$ for the mean would result in zero mean square error, while log-likelihood scores put greater emphasis on the determinant of the covariance and does not asses correlations. Therefore, we argue for the use of a much more direct method to assess the covariance. Specifically, we reason that the goal of estimating the covariance is to encode the relation between the target variables. *Therefore, partially observing a set of correlated targets should improve the prediction of the hidden targets since by definition the covariance encodes this correlation.* As an example, if $P$ and $Q$ are correlated, then observing $P$ should improve our estimate of $Q$. Hence, we propose a new metric that evaluates the accuracy of correlations which we call the *Conditional Mean Absolute Error* (CMAE) which is illustrated in Algorithm 1.

Formally, given an $n$-dimensional target prediction $\hat{y}$, ground truth $y$ and the predicted covariance $\text{Cov}(\hat{Y}|X{=}x)$, we define the Conditional Mean Absolute Error as $\sum_i |y_i - \tilde{y}_i|/n$, where $\tilde{y}_i$ is the updated mean obtained after conditioning $\mathcal{N}(\tilde{y}_i, \text{Cov}(\hat{Y}|X) \mid y_{j \neq i}, x)$. For each prediction $\hat{y}_i$, we obtain its revised estimate $\tilde{y}_i$ by conditioning it over the ground truth of the remaining variables $y_{i \neq j}$. This evaluation is reminiscent of *leave-one-out*, where we observe $\tilde{y}_i$ given other observations $y_{j \neq i}$. While *leave-one-out* can be generalized to *leave-k-out*, we do not observe any change in the evaluation trend. A method having lower *leave-one-out* also has a lower *leave-k-out* error. Moreover,

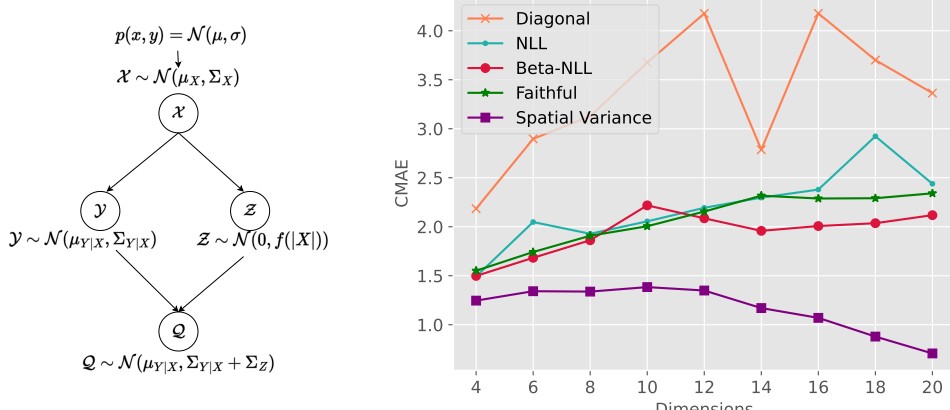

Figure 4: *Synthetic-Multivariate*. (Left) Experiment Design: We simulate correlated input-target variables with heteroscedastic covariance. (Right) Results: We plot the Conditional Mean Absolute Error (CMAE) metric (y-axis) for all methods from dimensions 4 to 20 (x-axis). We show that the gap between the Spatial Variance and other baselines widens with increasing dimensionality.

*leave-k-out* requires taking $\binom{n}{k}$ combinations which is significantly higher than taking $n$ combinations in *leave-one-out*. This motivates the use of *leave-one-out* strategy.

We measure the absolute error of this revised estimate against the ground truth of the unobserved variable and repeat for all $i$. An accurate estimate of $\mathrm{Cov}(\hat{Y}|X{=}\boldsymbol{x})$ will decrease the error whereas an incorrect estimate will cause an increase. We highlight that this metric is agnostic of downstream tasks involving covariance estimation. Hence, we use CMAE as a metric for all multivariate experiments.

## 5 EXPERIMENTS

The goal of this paper is accurate covariance estimation which is reflected in our experiments. Unlike previous literature, we specifically focus on multivariate *outputs*, which requires us to readdress several existing experimental designs. Our synthetic experiments consist of learning a univariate sinusoidal (inspired from Seitzer et al. (2022)) and a multivariate distribution. We conduct our real-world experiments on the UCI regression repository as well as on the MPII and LSP 2D human pose estimation datasets. Our baselines consist of the negative log-likelihood Dorta et al. (2018); Simpson et al. (2022); Gundavarapu et al. (2019) and its variations: the diagonal covariance (Kendall & Gal, 2017), $\beta$-NLL (Seitzer et al., 2022) and Faithful Heteroscedastic Regression (Stirn et al., 2023). We take special care to provide a fair comparison; all methods are initialized with the same initial mean and covariance estimates. Additionally, each method has its own optimizer and learning rate scheduler. Furthermore, the batching and ordering of samples is the same for all methods.

We conduct multiple trials for each experiment and report the mean only and not the standard deviation, since standard deviation is a measure of consistency and accounts for randomness for different runs in the *same evaluation setting*. However in our experiments we use *different settings* for each run. This change includes defining new splits for input-output variables (UCI regression) as well as new multivariate distributions (synthetic multivariate) *for each run*. As a result some input-output splits and distributions are significantly difficult to learn and therefore distort the standard deviation. In comparison, the mean value takes into account this difficulty since if the setting is difficult for one, it is difficult for all, and hence affects all the results similarly. Therefore we avoid using standard deviation which can be distorted by difficult splits/covariance initialization and instead report the mean across multiple runs.

### 5.1 SYNTHETIC DATA

***Univariate***. We repeat the experiments of Seitzer et al. (2022) with a major revision. First, we introduce heteroscedasticity and substantially increase the variance of the samples. Second, we

Table 2: *UCI Regression*. We compute ten trials over all the datasets and report the Conditional Mean Absolute Error. We show that the Spatial Variance outperforms all baselines on ten out of twelve datasets. While the Spatial Variance is not the best performing method on Naval and Parkinson, the value of CMAE is small and comparable to other baselines.

| Method | Abalone | Air | Appliances | Concrete | Electrical | Energy | Turbine | Naval | Parkinson | Power | Red Wine | White Wine |
|---|---|---|---|---|---|---|---|---|---|---|---|---|
| Diagonal | 5.49 | 8.03 | 11.71 | 7.86 | 10.06 | 7.12 | 7.07 | 4.31 | 8.56 | 8.16 | 7.96 | 8.44 |
| NLL | 3.28 | 3.42 | 2.41 | 4.16 | 7.14 | 5.10 | 3.40 | **0.18** | 1.86 | 6.22 | 5.81 | 7.26 |
| $\beta$-NLL Seitzer et al. (2022) | 2.85 | 5.67 | 4.89 | 7.21 | 8.41 | 6.17 | 5.03 | 1.15 | 5.48 | 6.73 | 6.96 | 7.08 |
| Faithful Stirn et al. (2023) | 2.96 | 3.27 | 1.79 | 3.93 | 7.36 | 2.90 | 3.29 | 0.20 | **1.68** | 5.81 | 5.74 | 6.89 |
| **Spatial Variance** | **1.83** | **2.27** | **1.39** | **2.82** | **4.89** | **2.34** | **2.40** | 0.28 | 2.54 | **3.87** | **4.05** | **4.60** |

simulate different sinusoidals having constant and varying amplitudes. We draw 50,000 samples and train a fully-connected network with Batch Normalization for 100 epochs.

Our experiments show that the negative log-likelihood and $\beta$-NLL fail to converge under noisy conditions. The negative log-likelihood incorrectly overestimates the variance due to the arbitrary mapping in the absence of supervision. The gradient updates in $\beta$-NLL are susceptible to large variances, which may negatively impact optimization, as shown in Fig. 3 (c). While Faithful Heteroscedastic Regression (FHR) Stirn et al. (2023) uses the mean squared error objective to achieve faster convergence, the resulting variance is incorrectly estimated. We theorize that this is because FHR trains the mean estimator assuming homoscedastic unit variance, whereas the variance estimator needs to model heteroscedasticity based on the homoscedastic assumption of the mean squared error.

***Multivariate***. We propose an additional synthetic data experiment for multivariate analysis to study heteroscedastic covariance. We let $X, Y$ be jointly distributed and sample $\boldsymbol{x}$ from this distribution. Subsequently, we sample $\boldsymbol{y}$ conditioned on $\boldsymbol{x}$. To simulate heteroscedasticity, we draw samples from $Z$, a new random variable whose covariance $\Sigma_Z = \text{diag}(\sqrt{|\boldsymbol{x}|})$ depends on $\boldsymbol{x}$. Since $Y$ and $Z$ are independent given $X$, their sum also satisfies the normal distribution $Q|X \sim \mathcal{N}(\mu_{Y|X}, \Sigma_{Y|X} + \Sigma_{Z|X})$. Therefore, the goal of this experiment is to model the mean and the heteroscedastic covariance of $Q$ conditioned on observations $\boldsymbol{x}$. The schematic for our experimental design is shown in Fig. 4.

For our experiments, we vary the dimensionality of $\boldsymbol{x}$ and $\boldsymbol{q}$ from 4 to 20 in steps of 2, and report the mean of ten trials for each dimension. We draw 4000 to 20000 samples and report our results using CMAE. We observe two trends in Fig. 4: first, as the dimensionality of the samples increases, the gap between the Spatial Variance and other methods widens. This is because with increasing dimensionality, the number of free parameters to estimate in the covariance matrix grows quadratically. An increase in parameters typically requires a non-linear growth in the number of samples for robust fitting. As a result, the difficulty of the mapping $\text{Cov}(\hat{Y}|X{=}\boldsymbol{x}) = g_\Theta(\boldsymbol{x})$ increases with dimensionality. Second, we observe the curious trend of the decrease in CMAE for the Spatial Variance as dimensionality increases. We believe this to be due to the fact that our ability to uniformly sample from high-dimensional spaces is limited, restricting the number of samples. Moreover, it is easier to fit few samples in high-dimensional spaces than fitting the same number of samples in low dimensions.

## 5.2 UCI REGRESSION

We perform our analysis on twelve multivariate UCI regression Dua & Graff (2017) datasets, which have been used in previous work on negative log-likelihood Stirn et al. (2023); Seitzer et al. (2022). However, the goal of this work is to study covariance estimation, which requires us to use different pre-processing since many of the datasets have univariate or low-dimensional targets. Specifically, for each dataset we randomly allocate 25% of the features as input and the remaining 75% features as multivariate targets at run-time. Indeed, some combinations of input variables may fare poorly at predicting the target variables. However, this is an interesting challenge for the covariance estimator, which needs to learn the underlying correlations even in unfavourable circumstances. Moreover, random splitting also allows our experiments to remain unbiased since we do not control the split of variables at any instant.

Since we are addressing *unsupervised* covariance estimation, we do not make training and evaluation splits of the dataset to increase the number of samples available for covariance estimation. While this may seem questionable, we reason that the covariance is a measure of correlation as well as variance.

Table 3: *2D Human Pose Estimation*. We compare the Spatial Variance using the Conditional Mean Absolute Error (CMAE) metric with other methods on the combined MPII and LSP/LSPET datasets. We show that the Spatial Variance can scale to convolutional architectures and outperform baselines.

| Method | head | neck | lsho | lelb | lwri | rsho | relb | rwri | lhip | lknee | lankl | rhip | rknee | rankl | **Avg** |
|---|---|---|---|---|---|---|---|---|---|---|---|---|---|---|---|
| MSE | 5.53 | 7.88 | 7.31 | 8.73 | 10.52 | 7.01 | 8.41 | 10.19 | 8.43 | 8.53 | 10.53 | 8.13 | 8.37 | 10.58 | 8.58 |
| Diagonal | 5.36 | 7.23 | 6.95 | 8.17 | 10.01 | 6.48 | 7.79 | 9.73 | 8.11 | 8.30 | 11.12 | 7.75 | 8.17 | 11.20 | 8.32 |
| NLL | 4.48 | 6.81 | 5.38 | 5.19 | 7.13 | 5.11 | 4.86 | 6.89 | 6.62 | 6.35 | 8.45 | 6.43 | 6.17 | 8.40 | 6.31 |
| $\beta$-NLL Seitzer et al. (2022) | 4.63 | 7.14 | 6.74 | 8.23 | 9.98 | 6.43 | 7.92 | 9.65 | 8.01 | 8.13 | 10.12 | 7.71 | 7.93 | 10.19 | 8.06 |
| Faithful Stirn et al. (2023) | 5.13 | 6.36 | 5.32 | 4.94 | 7.18 | 4.96 | 4.72 | 6.85 | 6.67 | 6.29 | 8.39 | 6.36 | 6.22 | 8.37 | 6.27 |
| **Spatial Variance** | **3.76** | **5.98** | **4.80** | **4.64** | **6.34** | **4.46** | **4.41** | **6.12** | **6.09** | **5.82** | **7.59** | **5.79** | **5.63** | **7.55** | **5.64** |

If too few samples are provided for training then the resulting conditional covariance $\text{Cov}(\hat{Y}|X{=}\boldsymbol{x})$ is nearly singular. Moreover, our evaluation continues to remain fair since covariance estimation is unsupervised and our experimental methodology is the same for all approaches. For all datasets, we follow the established machine learning practices of standardizing our variables with zero mean and a variance of ten (which allows better convergence for all methods). Standardizing the datasets also allows us to directly compare the CMAE across datasets. We perform 10 trials for each dataset and report our results in Table 2. The Spatial Variance outperforms all baselines on ten out of twelve datasets. Note that the CMAE is small across all methods for the remaining two datasets.

### 5.3 2D HUMAN POSE ESTIMATION

We introduce experiments on human pose (Kreiss et al., 2019; 2021; Newell et al., 2016; Liu & Ferrari, 2017; Shukla & Ahmed, 2021; Shukla, 2022; Yoo & Kweon, 2019; Shukla et al., 2022; Gong et al., 2022) particularly because of the challenges it poses to modeling the Spatial Variance. Popular human pose architectures such as the hourglass (Newell et al., 2016) are fully convolutional, whereas all our previous experiments have focused on fully-connected architectures. Additionally, while the Spatial Variance assumes $\boldsymbol{x} \in \mathbb{R}^m$ and $\boldsymbol{y} \in \mathbb{R}^n$, human pose estimation relies on input images $\mathsf{X} \in \mathbb{R}^{C \times H \times W}$ and output heatmaps $\mathsf{Y} \in \mathbb{R}^{\#\text{joints} \times 64 \times 64}$. However, we show that the Spatial Variance outperforms all baselines on human pose with just a few modifications. The first modification is the use of soft-argmax (Li et al., 2021b;a), which reduces the output heatmap to a keypoint vector $\boldsymbol{y} \in \mathbb{R}^{\#\text{joints}*2}$. The second modification recursively calls the hourglass module till we obtain a one-dimensional vector encoding which serves as the input for our method.

We use the popular Stacked Hourglass (Newell et al., 2016) as our backbone for human pose estimation. We run our experiments on two popular single person datasets: MPII (Andriluka et al., 2014) and Leeds Sports Pose (LSP-LSPET) (Johnson & Everingham, 2010; 2011). We perform our analysis by merging the MPII and LSP-LSPET datasets to increase the number of samples. We continue to use CAME as our metric since for single person estimation, the scale of the person is fixed and hence CMAE is highly correlated to PCKh/PCK (the preferred metric for multi-person multi-scale pose estimation). Moreover, the Euclidean distance as a measure of error is also used in MPJPE, which is another well-known evaluation metric for human pose. We perform five trials and report our results in Table 3. Our experiments show that the Spatial Variance outperforms all baselines and is successfully able to scale to convolutional architectures.

## 6 CONCLUSION

This paper studied unsupervised heteroscedastic covariance estimation through parametric neural networks. We addressed a key limitation in negative log-likelihood; in the absence of supervision, $\text{Cov}(\hat{Y}|X{=}\boldsymbol{x}) = g_{\Theta}(\boldsymbol{x})$ is essentially an arbitrary mapping of $\boldsymbol{x}$ to a positive definite matrix which may not represent the randomness in $\hat{\boldsymbol{y}}$. Our solution, the Spatial Variance, is a novel derivation of a closed-form expression for $\text{Cov}(\hat{Y}|X{=}\boldsymbol{x})$ through its Taylor polynomial. Doing so allowed us to represent the variation in $\hat{\boldsymbol{y}}$ through its gradient and curvature. Additionally, we addressed the lack of direct methods to evaluate the covariance by proposing the Conditional Mean Absolute Error (CMAE) metric. The metric uses conditioning of the normal distribution to quantify the accuracy of learnt correlations. We performed extensive experiments and show that the Spatial Variance outperforms all baselines implying a better ability to learn covariances in an unsupervised framework.

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
