# OpenReview forum: "How To Train Your Covariance"
_ICLR.cc/2024/Conference — Submitted to ICLR 2024_

### Official Review · Reviewer_Q3zL · 2023-10-12

**Soundness:** 1 poor
**Presentation:** 2 fair
**Contribution:** 1 poor
**Rating:** 1
**Confidence:** 5

**Summary:**

The paper studies a conditional covariance estimation problem, where the covariance can vary depending on the conditioned random variable $x$. The paper points out that using NLL can be problematic and propose an alternative formulation and a metric. The paper applies the proposed scheme on various datasets and compares with other baselines.

**Strengths:**

- Related papers are thoroughly reviewed.

- Experiment is comprehensive and covers various datasets.

**Weaknesses:**

- Although the paper studies a theoretical problem (conditional covariance estimation), a precise probabilistic statement of the problem is nowhere provided. I was confused starting from the first paragraph, which says ``$p(y | x)$ follows $N(y, \Sigma_{y | x})$''. Isn't $y$ a random variable? How can a random variable be a conditional mean of itself? Not only in this paragraph, but in many other places throughout the paper, the notations $y, \hat{y}, f(x), f_\theta(x)$ were carelessly used, making the paper almost impossible to understand. I believe the paper can be much improved by clearly defining the problem (under which probability distribution the data is generated, which parameter is being estimated, what assumptions are used, etc.).

- Most equations in the paper remain at the level of heuristic. While it is okay to have a heuristic explanation of the proposed method, the way it is presented in the current draft is unnecessarily confusing. For example, how do we even define the limit in Eq. (1)? I think the paper should try to minimize the use of non-rigorous math.

**Questions:**

- To my knowledge, conditional mean/covariance estimation is impossible without further assumption (e.g. regularity of $\mu_{y | x}, \Sigma_{y | x}$). I wonder how the authors avoided this problem.

- Most figures are missing axis labels. For example, Figure 1 is placed in the very beginning of paper without explaining what the curves are.

---

> ### Author Response · Authors · 2023-11-15
>
> We thank you for your review. We would like to clarify below:
>
> 1. **To my knowledge, conditional mean/covariance estimation is impossible without further assumption.**\
> Conditional mean/covariance estimation has been studied in multiple previous works, e.g.: [1-5]. The common approach in these works is to assume that the target distribution $p(Y | X = x)$  conditioned on the input $x$ (can also be an image) follows a normal distribution $\mathcal{N}(\mu_{Y|X}, \Sigma_{Y|X})$. The mean and covariance of the target are predicted using two neural networks which takes $x$ as input and are trained using the negative log-likelihood. Our work follows  this paradigm, and proposes a new way to model the covariance. Therefore, it is not impossible to assume conditional mean/covariance estimation within our problem statement.
>
> 1. **Most equations in the paper remain at the level of heuristic... how do we even define the limit in Eq. (1) ...**\
> The limit in Eq. 1 is a standard definition for continuous distributions, and we have added references for the same. The only heuristic is in treating the limit as a random variable with zero mean, the variance of which is learnt through neural network optimization. We take a principled approach in deriving our closed-form expression for the covariance, and fail to understand the reviewer's concern.
>
> 1. **Although the paper studies a theoretical problem ...** \
> We have addressed this point by rearranging the content of the paper. The notation table has been moved from the related work section to the introduction. We have moved the negative log-likelihood equation from the related work to the introduction, since we optimize our covariance formulation through the negative log-likelihood. Our introduction already stated our assumption that the target distribution is a multivariate normal distribution, the standard assumption in machine learning. Furthermore, we also specify the standard assumption that we do not know the data generating distribution p(X, Y). Instead, we obtain samples (x, y) from it to form our dataset. We understand that our notation may have been ambiguous, and we have revised the draft to address this.
>
> 1. **Most figures are missing axis labels ...**\
> Thank you for pointing this out. We have addressed this in the **revised draft.**
>
> We would like to conclude the response by reiterating that this work is in line with previous works which propose models and training strategies to learn the covariance in applied settings.
>
>
> [1] Andrew Stirn, Harm Wessels, Megan Schertzer, Laura Pereira, Neville Sanjana, and David Knowles.
> Faithful heteroscedastic regression with neural networks. In International Conference on Artificial
> Intelligence and Statistics, pp. 5593–5613. PMLR, 2023
>
> [2] Maximilian Seitzer, Arash Tavakoli, Dimitrije Antic, and Georg Martius. On the pitfalls of het-
> eroscedastic uncertainty estimation with probabilistic neural networks. In International Confer-
> ence on Learning Representations, 2022
>
> [3] Nitesh B Gundavarapu, Divyansh Srivastava, Rahul Mitra, Abhishek Sharma, and Arjun Jain. Struc-
> tured aleatoric uncertainty in human pose estimation. In CVPR Workshops 2019.
>
> [4] Ivor JA Simpson, Sara Vicente, and Neill DF Campbell. Learning structured gaussians to approximate
> deep ensembles. In Proceedings of the IEEE/CVF Conference on Computer Vision and Pattern
> Recognition 2022
>
> [5] Garoe Dorta, Sara Vicente, Lourdes Agapito, Neill DF Campbell, and Ivor Simpson. Structured
> uncertainty prediction networks. In Proceedings of the IEEE conference on computer vision and
> pattern recognition 2018.

---

> > ### Comment · Reviewer_Q3zL · 2023-11-17
> >
> > Thank you for the response. I decided to keep my score for the following reasons.
> >
> > **1. To my knowledge, conditional mean/covariance estimation is impossible without further assumption.**
> >
> > In nonparametric statistics literature, it is standard to have a certain regularity condition (Holder or Sobolev condition) on the mean/covariance function. See e.g. https://arxiv.org/pdf/1202.5134.pdf and references therein. Without this assumption, estimating the mean/covariance function is simply impossible. To see this point, let's say we have a finite sample $(x_1, y_1), \dots, (x_n, y_n)$ from a continuous distribution $p_{X, Y}$. Then, with probability 1, none of $x_i$ will appear more than once, so $y_i$ is the only information we know about the conditional distribution $p_{Y | X = x_i}$. Therefore, unless we assume some regularity, mean/covariance function estimation is a statistically ill-posed problem. I would suggest to (1) clarify the condition under which the mean/covariance estimation is possible (2) discuss why neural network is a good choice to model those functions.
> >
> > **2. Most equations in the paper remain at the level of heuristic.**
> >
> > Eq. (2) in the revision is still a mathematically nonsense expression. Please clarify which part of the added reference I can refer to. I cannot find an immediate way to justify such an expression in standard measure-theoretic probability theory.
> >
> > **3. Although the paper studies a theoretical problem..**
> >
> > Despite the changes made, many of the notations remain unclear and confusing. The problem needs to have a clearer statistical description, and the notations $y, \hat{y}, f(x), f_\theta(x)$ have to be carefully distinguished.

---

> ### Author Response · Authors · 2023-11-21
>
> Dear Reviewer,
>
> Our work builds upon a wide range of successful parametric variance / covariance estimation methods. This includes the highly cited: Kendall and Gal [NeurIPS ‘17], Dorta et al. [CVPR ‘18], Seitzer et al. [ICLR ‘22], Stirn et al. [AISTATS ‘23]. Many more works use parametric covariance estimation for downstream tasks, e.g.: Lu and Koniusz [CVPR ‘22], Liu et al. [ICRA ‘18], Gundavarapu et al. [CVPRW ‘20], Russel and Reale [T-NNLS ‘21], Simpson et al. [CVPR ‘22] among many others. These references are available in the draft. We follow and describe a similar (and _same_  in some cases) probabilistic framework, and maintain standard machine learning notation which is explicitly described through a table.
>
> We therefore confess that we are perplexed by the evaluation of the draft by the reviewer.
>
> 1. **In nonparametric statistics literature, it is standard ...** \
> Neural networks with differentiable activation functions and optimized using negative log-likelihood with Tikhonov regularization are assumed to be Lipschitz and smooth [6]. These assumptions are implied and well known in deep learning research, if needed we can add them in the draft. Additionally, results from the theory of neural networks highlight that neural networks with non-constant, bounded, continuous activation functions, with continuous derivatives up to order K belong to Sobolev spaces of order K [7]. Further, the use of neural networks to model the variance / covariance has already been explored in previous work on parametric covariance estimation, hence we do not emphasize this in the draft. The biggest advantage of neural networks is that it allows us to learn complex representations from varying kinds of inputs (such as images) to predict the covariance.
>
> 1. **Eq. (2) in the revision is still a mathematically nonsense expression...** \
> Let us assume we have $N$ predictions $(x, \hat{y})^{(1)} \ldots (x, \hat{y})^{(N)}$. However, the probability of exactly observing $p(X = x)$ is zero for continuous variables. Instead, the standard approach (Section 2.4 in [8]) is to observe over the set $X \in lim\_{\varepsilon \rightarrow 0} [x, x + \varepsilon]$. This implies that for continuous variables, we do not observe a specific but a range of values around $X = x$. We can incorporate this definition in the covariance too. Since $\hat{Y}$ is a deterministic transformation of $X$ through a parametric network, $\hat{Y} = f\_{\theta}(X)$, we have: $Cov(\hat{Y} | X = x) = Cov( f\_{\theta}(X) | X \in lim\_{\varepsilon \rightarrow 0} [x, x + \epsilon])$. \
> Finally, the only heuristic in our derivation is in treating  this neighborhood $\lim\_{\varepsilon \rightarrow 0} [x, x + \varepsilon]$ as $x + \epsilon$, where $\epsilon$ is an m-dimensional random variable with a zero-mean isotropic Gaussian distribution $p(\epsilon) = \mathcal{N}(0, \sigma^2\_{\epsilon}(x) I_m)$. Introducing $\epsilon$ imposes a distribution on the neighborhood which is centred at $x$. This also allows us to represent $\hat{y} = f\_{\theta}(x + \epsilon)$ stochastically. While the variance of this random variable is unknown (we later show that it can be learnt), we assume heteroscedasticity which allows us to represent neighborhoods of varying spatial extents for each $x$.
> We elaborate on expression 2 in subsection 3.1 of the revised draft. \
> We take a principled approach in deriving our closed-form expression for the covariance, which we believe is not reflected in the current review.
>
> 1. **Despite the changes made, many of the notations remain unclear and confusing ...** \
> We already follow standard machine learning conventions for $y, \hat{y}, f\_{\theta}$ and described them in Table 1. Further, our introduction also stated that $p(Y | X)$ is normally distributed, and parameterized by two networks $f\_{\theta}$ and $g\_{\Theta}$, which are trained through the negative log-likelihood. We emphasized that our work assumes the same probabilistic setting as prior literature on parametric covariance estimation, and the only change we propose is in the way we model the covariance through a novel closed form expression. Without specific examples we are not able to understand what in the notation is confusing.
>
> Nevertheless, we thank you for your review.
>
> [6] Du, S., Lee, J., Li, H., Wang, L. and Zhai, X., 2019, May. Gradient descent finds global minima of deep neural networks. In International conference on machine learning (pp. 1675-1685). PMLR.
>
> [7] Czarnecki, W.M., Osindero, S., Jaderberg, M., Swirszcz, G. and Pascanu, R., 2017. Sobolev training for neural networks. Advances in neural information processing systems, 30.
>
> [8] Evans, Michael J., and Jeffrey S. Rosenthal. Probability and statistics: The science of uncertainty. Macmillan, 2004.

---

### Official Review · Reviewer_kqRb · 2023-10-28

**Soundness:** 3 good
**Presentation:** 3 good
**Contribution:** 3 good
**Rating:** 6
**Confidence:** 4

**Summary:**

This paper tackles the problem of unsupervised covariance estimation when the covariance is not homogeneous across samples. Current solution use neural networks with negative log-likelihood objectives. They show that the obtained solutions for the covariance do not take into account the randomness in the mean estimation. To tackle that, they propose a solution that capture the randomness in the mean by incorporating local curvature around the samples. Furthermore, they propose an evaluation metric Conditional Mean Absolute Error (C-MAE) to quantify the covariance estimation in the absence of annotations.

**Strengths:**

The paper tackles an important practical problem for statistical machine learning. The proposed spatial variance motivated by taking into curvature around samples is a nice approach to account for uncertainty in mean and covariance estimation. The new proposed metric C-MAE would also be useful in other applications involving statistics estimations in unsupervised settings as an alternative to log-likelihood.

**Weaknesses:**

The computational complexity of the approach could be an issue as the paper tackles a practical estimation problem. The paper does not compare to other approaches to log-likelihood such as Lotfi, S., Izmailov, P., Benton, G., Goldblum, M., & Wilson, A. G. (2022, June). Bayesian model selection, the marginal likelihood, and generalization. In International Conference on Machine Learning (pp. 14223-14247). PMLR.

**Questions:**

- In Section 2.1, \sigma_{\Theta} has not been defined. Is it a scalar that is assuming a diagonal covariance matrix?, How do you go from Cov(\hat{y}) to \sigma_{\Theta} ?
- The sentence after Equation (6) is incomplete: "We note that both both Cov() and Cov()..."
- What is the theoretical explanation motivating the use of the thirs matrix term k_3(x). It is said in the paper, that the curvature of the function at x cannot alone explain the stochasticity of the samples which motivate the use of k_3(x). It would then be appropriate to motivate the definition of "spatial covariance" by the the "curvature of x" and of "?" coming from k_3(x). Could the authors please elaborate more on this?
-The following paper proposes "Conditional Marginal likelihood" as an alternative to the likehood (although for generalisation context): Lotfi, S., Izmailov, P., Benton, G., Goldblum, M., & Wilson, A. G. (2022, June). Bayesian model selection, the marginal likelihood, and generalization. In International Conference on Machine Learning (pp. 14223-14247). PMLR. This relates to your definition of C-MAE. Could you elaborate on the differences between the two metrics and possibly compare them in the current setting?
- The computational complexity of the approach could be an issue as the paper tackles a practical estimation problem. Could the authors please provide an exact analysis of the computational complexity of the method and suggest possible ways of improvement?

---

> ### Author Response · Authors · 2023-11-15
>
> We thank you for your review. We address your questions below:
>
> 1. **In Section 2.1, $\sigma_{\Theta}$ has not been defined ...** \
> We have clarified this in the related work in the **revised draft**. $\sigma_{\Theta}$ (or Var$_\theta$) is a vector encoding a diagonal covariance matrix. The authors of Beta-NLL specify Cov($\hat{y}$) to be a diagonal matrix.
> 1. **The sentence after Equation (6)** \
> We have completed the sentence.
> 1. **What is the theoretical explanation motivating the use of the thirs matrix term $k_3(x)$ ...** \
> We have addressed this in the revised draft. The sum of two independent Gaussians is the sum of the means and covariances of the two Gaussians. Therefore, the covariance of a sample can be attributed to the nature of the function underlying the sample as well as independent stochasticity, which is not a function of the input. We attributed $k_1(x)$ and $k_2 (x)$ to the former and $k_3(x)$ to the latter. Given the independence assumption, the final formulation can be written as the sum of all three terms.
> 1. **Conditional Marginal Likelihood comparison.** \
> Thank you for pointing us to this paper, which we have included in our related work. However, we believe that not comparing to Lotfi et al. should not be considered a weakness since the two works are fundamentally different. Conditional Marginal Likelihood is a metric to evaluate generalization. The metric does so by creating two non-overlapping subsets of the dataset: $D_1$ and $D_2$, training a model on $D_1$, and quantifying its performance on $D_2$. By contrast, CMAE is a metric with a different objective. Given a sample $(x, y)$ with its prediction $(\hat{y}, \textrm{Cov}(\hat{y}))$, the metric evaluates the covariance by measuring the improvement in $\hat{y}$ when $y$ is partially observed. Hence, the two metrics are fundamentally different.
> 1. **Computational complexity ...** \
> Indeed, we note in our limitations section that the primary bottleneck of our approach is in computing the Hessian. While determining the computational complexity of the Hessian for a generalized network is non-trivial, we draw your attention to [1]. Computing the Hessian for a function that maps an m-dimensional x to an n-dimensional y has a complexity of $O(nm^3)$.
> In practice, covariance estimation can be performed using a smaller, proxy model in place of a large model (which could be retained for mean estimation). The reduced parameter count would decrease the computational requirements of computing the Hessian.
>
>
>
> [1] Yao, Zhewei, et al. "Pyhessian: Neural networks through the lens of the hessian." 2020 IEEE international conference on big data (Big data). IEEE, 2020.

---

> > ### Comment · Reviewer_kqRb · 2023-11-20
> >
> > I thank the authors for modifying the manuscript and for providing some details about the unclear parts.

---

> > > ### Author Response · Authors · 2023-11-22
> > > **Discussion Period Ends Soon**
> > >
> > > Dear Reviewer kqRb,\
> > > We thank you for your time, efforts and your reply!\
> > > We believe that our revised manuscript and response addressed your questions, and we would be grateful if you would consider accepting the manuscript.\
> > > Once again, thank you for your time and efforts!

---

### Official Review · Reviewer_r64o · 2023-10-30

**Soundness:** 2 fair
**Presentation:** 2 fair
**Contribution:** 2 fair
**Rating:** 5
**Confidence:** 4

**Summary:**

The authors propose a novel approach to train covariances, stemming from the observation that, when learning Gaussian models, the mean and the covariance are independently parametrised but they affect one another during training. Consequently, they define a new parametrisation for the covariance that is *tied* to the curvature of the mean function around its argument. The approach is compared to ML and other recent methods using synthetic and real-world data, such comparison is performed under a novel performance indicator for covariance modelling introduced in this paper too.

**Strengths:**

The idea of questioning the standard approach to training covariances is undoubtedly of interest to the community. Furthermore, developing this new approach and a performance indicator is a valuable contribution.

**Weaknesses:**

Although the general idea is attractive and, to some extent, promising, the concept is not properly exploited in the article. In this regard, the most relevant weaknesses of the paper are:

- Generally, the paper could be clearer, and its format can be improved. For instance:
    - the abstract (5th line) defines the mean of a Gaussian as $f(x)$, and the covariance as $Cov(f(x))$. So, it is not clear whether $f(x)$ refers to the RV to be modelled or its mean.
    - Figs 1 and 2 are not referred to in the body of the paper, and they are not self-explanatory either. To this Reviewer, their purpose is not clear.
    - Tables span beyond the margins of the text
    - A few times, it is mentioned that the experiments are run over _multiple network architectures_; however, in the experiments, there is no mention of specific architectures used
    - gaussian, hessian -> Gaussian, Hessian
    - axis labels in Fig 3 are too small

- Also, in the line of clarity, the paper is motivated by the pitfalls of maximum likelihood (ML). However, the proposal in the paper results in a specific parametrisation of the covariance, which ties the structure of the covariance and the mean (eq 8 shows how the covariance contains the Jacobian of the mean function). Therefore, the proposal is not _another training strategy_ but rather a covariance parametrisation. As a matter of fact, after eq 8 the authors state that their parametrisation is used alongside ML.

- Benchmarks are unclear: The experiments compare the proposed parametrisation against **NLL** (though the proposed method also uses NLL as far as I understand), **Diagonal**, which I assume also uses NLL and other methods. It is thus confusing if the paper compares approaches to training covariances or models for covariances.

- Another point worth noticing is the fact that the paper proposes a variance parametrisation and also a performance indicator (C-MAE). However, this is the only performance indicator used in the experiments, meaning that other than the conceptual justification of C-MAE (which I find valid), there is no experimental validation. This means that the authors propose a model and use their own defined metric to assess it.

- There should be given more details about the choice of $k_1,k_2,k_3$, the networks, and the learning objectives.

**Questions:**

Please refer to the comments in the previous section

---

> ### Author Response · Authors · 2023-11-15
>
> We thank you for your review. We understand that the concerns majorly revolve around presentation and formatting. We have addressed the figures, tables and capitalisation in the **revised draft**. We also answer your specific questions:
>
> 1. **So, it is not clear whether  refers to the RV to be modelled or its mean.** \
> $f(x)$ refers to the mean of the random variable being modelled. We have updated this in the abstract.
> 1. **...multiple network architectures...** \
> The architecture used depends on the task being solved. For our synthetic experiments and UCI Regression, we use fully connected networks to learn $y$ in $R^m$ from $x$ in $R^n$. We specifically choose the task of human pose estimation to showcase the use of convolutional neural networks, and regress heatmaps of shape $R^{j \times 64 \times 64}$ from images of shape $R^{256 \times 256}$. Collectively, our experiments address different shapes and types of input-target pairs. We have rephrased this sentence in the draft.
> 1. **Also, in line of clarity ...** \
> We agree with your observations. The proposal is indeed a way to model the covariance which is learnt via the negative log-likelihood. To avoid ambiguity, we have renamed the paper from "_How To Train Your Covariance_" to "_How To Model Your Covariance_". We also rephrased the text to address this ambiguity wherever applicable in the draft.
> 1. **Benchmarks are unclear.** \
> Indeed, we compare our method against other approaches to model and train the covariance. While NLL, Diagonal and our proposed model for the covariance use NLL as a training objective, Beta-NLL and Faithful Heteroscedastic Regression have their own training objectives, which are variants of NLL. All these models/training strategies give a covariance prediction with which we compare our proposed method.
> 1. **Another point worth noticing ...** \
> While we understand your concern, we believe that this point should not be considered as a weakness. The reason we propose a new metric is because of the lack of any measure to directly assess the covariance. Moreover, CMAE in our opinion elegantly combines the notion of L1/L2 error with correlations through the fundamentals of conditioning a distribution. Additionally, the metric is independent of the training objective for all methods, and hence we believe it is a fair comparison. We hope that this paper sparks discussion in the community not only on covariance estimation but more importantly on covariance evaluation.
> 1. **There should be more details about $k_1, k_2, k_3$ ...** \
> We have addressed this towards the end of our methodology section.  The covariance estimator $g_{\Theta}(x)$ predicts $k_1(x), k_2(x)$ and $k_3(x)$, where $k_1(x), k_2(x)$ are positive scalars. We enforce $k_3(x)$ to be positive semi-definite by predicting an unconstrained matrix and multiplying it with its transpose, similar to previous work.

---

> ### Author Response · Authors · 2023-11-22
> **Discussion Period Ends Soon**
>
> Dear Reviewer r64o,\
> We thank you for your time and efforts. \
> Since the discussion period ends soon, please let us know if we could resolve your questions.\
> We believe that our revised draft addresses your concerns.\
> We welcome you to read our general response as well.\
> If satisfied, we would be grateful if you would consider accepting the manuscript.\
> Once again, thank you for your time and efforts!

---

### Official Review · Reviewer_G5cy · 2023-11-01

**Soundness:** 3 good
**Presentation:** 3 good
**Contribution:** 3 good
**Rating:** 6
**Confidence:** 3

**Summary:**

The paper proposes a new method to lean the multivariate target distribution, namely the mean and heteroscedastic covariance. The authors contributions are two folds: 1) a concept of spatial variance by studying the curvature around input x. 2) conditional mean absolute error for evaluations.

**Strengths:**

Overall I believe the paper is well motivated, and new concepts such as spatial variance are clearly presented. The experiment sections are also thorough to demonstrate the effectiveness of the proposed methods.

Originality: Two key concepts presented in this paper (spatial variance and C-MAE) are novel.

Quality: The paper is written with good quality. The authors motivated the problem well, provided detailed derivations and extensive experiment results.

Clarity: The paper is easy to follow.

Significant: The paper is important as it provides a new method for heteroscedastic covariance learning.

**Weaknesses:**

See questions below.

**Questions:**

I mainly have the following several questions:

The C-MAE operates like a leave one out fashion. In general for multi-variate Gaussian, we can do any leave-k-out. Would it provide more info if the consider any k greater than 1? Or mainly, the authors may want to illustrate the specific choice of leave one out here.
What are the specific choice considerations for k1, k2 and k3? Would any regularity conditions further help with supervision?

---

> ### Author Response · Authors · 2023-11-15
>
> We thank you for your review. We address your questions below:
>
> 1. **leave-one-out vs leave-k-out** \
> We have updated the CMAE section in the **revised draft** to address this. While _leave-one-out_ can be generalized to _leave-k-out_, we do not observe any change in the evaluation trend. A method having a lower _leave-one-out_ error also has a lower _leave-k-out_ one. Moreover, _leave-k-out_ requires taking nCk combinations, i.e., significantly more than the $n$ combinations required in _leave-one-out_. This motivates the use of the _leave-one-out_ strategy.
> 1. **What are the specific choices for $k_1, k_2, k_3$?** \
> We have addressed this towards the end of our methodology in our revised draft.  The covariance estimator $g_{\Theta}(x)$ predicts $k_1(x), k_2(x)$ and $k_3(x)$, where $k_1(x), k_2(x)$ are positive scalars. We enforce $k_3(x)$ to be positive semi-definite by predicting an unconstrained matrix and multiplying it with its transpose, similar to previous work.
> 1. **Would any regularity conditions further help with supervision?** \
> We believe that having priors would allow for faster and better convergence of the covariance. To maintain parity with previous work, we do not assume priors. Moreover, we work within the framework of unsupervised heteroscedastic covariance estimation, which means that we do not have supervisory signal for the covariance.

---

> > ### Author Response · Authors · 2023-11-22
> > **Discussion Period Ends Soon**
> >
> > Dear Reviewer G5cy,\
> > We thank you for your time and efforts. \
> > Since the discussion period ends soon, please let us know if we could resolve your questions.\
> > We welcome you to read our general response as well as look at our revised draft.\
> > If satisfied, we would be grateful if you would consider accepting the manuscript.\
> > Once again, thank you for your time and efforts!

---

### Author Response · Authors · 2023-11-21
**General Response And Revision Details**

Dear Reviewers,

We thank you for the time and effort taken to review the draft.

We are glad that the reviewers identified various strengths as well as potential weaknesses in the submission. Specifically, while noting that the Spatial Variance and CMAE are valuable contributions, reviewer **r64o** identified that the formatting of the paper can be significantly improved. Reviewer **G5cy** acknowledged the originality and significance of the paper, while wanting clarifications regarding certain aspects of CMAE and the Spatial Variance. The paper was also well received by reviewer **kqRb**, who noted the computational complexity of our approach as well as requested a comparison with Lotfi et al. While we have answered each review individually, we wanted to summarize the major changes we made to the draft. **We note that these changes focus on improving the presentation and writing; our results and formulation are unchanged.**

Specifically, reviewer **r64o** noted that the Spatial Variance is a new model and not a strategy for covariance estimation. We agree with this observation and propose renaming our paper from _“How To Train Your Covariance”_ to _“How To Model Your Covariance.”_ Additionally, we have improved the general presentation of the paper including fixing the figures and table margins. Reviewers **r64o, G5cy** and **kqRb** requested additional details regarding $k_1(x), k_2(x), k_3(x)$, which we have provided at the end of the methodology section. Reviewer **G5cy** asked for a discussion on leave-one-out vs leave-k-out, which we have included in the CMAE section.

We address reviewer **Q3zL**’s comments by improving the presentation of our math. We moved the notations table and the negative log-likelihood equation to the introduction, clearly specifying the objective that we are optimizing for in the introduction. We redefine our notation to prevent ambiguity wherever applicable. We dedicate a new subsection in the methodology to clarify Eq. (2). Additionally, we also point the reviewer to significant previous works in parametric covariance estimation since our style and notation is influenced by these works. While we disagree with the reviewer on certain points, we hope to work with the reviewer further to remove any ambiguities.

Finally, we would like to emphasize that this paper is aimed at applied covariance estimation through machine learning. While we do not present theoretical guarantees, we provide an experimentally proven formulation of the covariance derived through a first principles approach. We thank the reviewers once again for their time.

---

### Meta-Review · Area_Chair_Pa9S · 2023-12-07

**Metareview:**

This paper studies unsupervised heteroscedastic covariance estimation, where the goal is to learn the covariance of a multivariate target distribution without annotated data. The study introduces Spatial Variance as a way to capture the randomness in covariance by considering its curvature around the mean. Additionally, it introduces the Conditional Mean Absolute Error (C-MAE) as a performance metric in the absence of ground-truth data. The experiments demonstrate that the proposed approach outperforms existing methods on various datasets and network architectures, effectively modeling the relationships between target random variables.


The paper underwent a comprehensive review by four evaluators. While the majority of the reviewers acknowledged the paper's innovation in introducing the Spatial Variance and the novel C-MAE metric, concerns were raised by two reviewers. One reviewer expressed reservations about the authors relying primarily on their own metric for validation, considering it an insufficient basis for experimental assessment. Another reviewer took a more critical stance, questioning the rigor and clarity of the mathematical arguments presented in the paper. I concur with this latter viewpoint and suggest that the paper would significantly benefit from a more rigorous analysis of the proposed methodology. For instance, there is a recurring use of approximations in several equations throughout the paper, such as the application of Taylor expansion in Eq. 4, which is also found in Eqs 8 and 9, yet these approximations are not adequately characterized in terms of their approximation errors. Consequently, I firmly believe that the paper necessitates a substantial revision to address these concerns and enhance its overall quality.

**Justification For Why Not Higher Score:**

The paper's mathematical rigor is notably lacking, and it is imperative to address this issue for the paper to be considered for acceptance.

**Justification For Why Not Lower Score:**

N/A

---

### Decision · Program_Chairs · 2024-01-16

Reject